# Reported COVID-19 Vaccination Coverage and Associated Factors among Members of Athens Medical Association: Results from a Cross-Sectional Study

**DOI:** 10.3390/vaccines9101134

**Published:** 2021-10-04

**Authors:** Georgios Marinos, Dimitris Lamprinos, Panagiotis Georgakopoulos, Georgios Patoulis, Georgia Vogiatzi, Christos Damaskos, Anastasia Papaioannou, Anastasia Sofroni, Theodoros Pouletidis, Dimitrios Papagiannis, Emmanouil K. Symvoulakis, Kostas Konstantopoulos, Georgios Rachiotis

**Affiliations:** 1Department of Hygiene, Epidemiology and Medical Statistics, School of Medicine, National and Kapodistrian University of Athens, 11527 Athens, Greece; 2Emergency Department, Laiko General Hospital, 11527 Athens, Greece; dimitrislamprinos@gmail.com (D.L.); panos.k.georgakopoulos@gmail.com (P.G.); a.sofroni@hotmail.com (A.S.); tpouletidis@gmail.com (T.P.); 3Athens’s Medical Association, 11526 Athens, Greece; gipattt@gmail.com; 41st Cardiology Department, Medical School, National & Kapodistrian University of Athens, Hippokration Hospital, 11527 Athens, Greece; gvogiatz@yahoo.gr; 5N.S. Christeas Laboratory of Experimental Surgery and Surgical Research, Medical School, National and Kapodistrian University of Athens, 11527 Athens, Greece; x_damaskos@yahoo.gr; 6Health Center of Nea Makri, 19005 Attica, Greece; anpapai@yahoo.com; 7Public Health & Vaccines Laboratory, Department of Nursing, School of Health Science, University of Thessaly, 41110 Larissa, Greece; dpapajon@gmail.com; 8Clinic of Social and Family Medicine, Faculty of Medicine, University of Crete, 71003 Heraklion, Greece; symvouman@yahoo.com; 9Department of Haematology and Bone Marrow Transplantation Unit, National and Kapodistrian University of Athens, School of Medicine, Laikon General Hospital, 11527 Athens, Greece; kkonstan@med.uoa.gr; 10Department of Hygiene and Epidemiology, Faculty of Medicine, University of Thessaly, 41222 Larissa, Greece; grach@uth.gr

**Keywords:** COVID-19, vaccination coverage, vaccines hesitancy, health professionals

## Abstract

There are limited data on the prevalence and determinants of COVID-19 vaccination coverage among physicians. A cross-sectional, questionnaire-based, online study was conducted among the members of the Athens Medical Association (I.S.A.) over the period 25 February to 13 March 2021. All members of I.S.A. were invited to participate in the anonymous online survey. A structured, anonymous questionnaire was used. Overall, 1993 physicians participated in the survey. The reported vaccination coverage was 85.3%. The main reasons of no vaccination were pending vaccination appointment followed by safety concerns. Participants being informed about the COVID-19 vaccines by social media resulted in lower COVID-19 vaccination coverage than health workers being informed by other sources. Logistic regression analysis demonstrated that no fear over COVID-19 vaccination-related side effects, history of influenza vaccination for flu season 2020–2021, and the perception that the information on COVID-19 vaccination from the national public health authorities is reliable, were independent factors of reported COVID-19 vaccination coverage. Our results demonstrate a considerable improvement of the COVID-19 vaccination uptake among Greek physicians. The finding that participants reported high reliability of the information related to COVID-19 vaccination provided by the Greek public health authorities is an opportunity which should be broadly exploited by policymakers in order to combat vaccination hesitancy, and further improve COVID-19 vaccination uptake and coverage among physicians/HCWs, and the general population.

## 1. Introduction

Vaccines and vaccination are considered to be among the greatest public health achievements of the 20th century [1]. Undoubtedly, vaccinations have brought tremendous benefits to individuals, populations, health and economy [2]. However, along with the increased use of vaccines in public health, there are also growing public concerns about vaccine safety [3]. Ιt is well known that concern over vaccine safety is the most critical factor of vaccine hesitancy for both health care workers (HCWs) and the general public [4]. This phenomenon is called “vaccine hesitancy”. In particular, the Strategic Advisory Group of Experts (SAGE) on immunization, charged with advising WHO on vaccination, defined vaccine hesitancy as “a behavior, influenced by a number of factors including issues of confidence (level of trust in vaccine or provider), complacency (perceived risks of vaccine-preventable diseases are low), and convenience (access issues)” [5,6]. The protection and immunization of health care workers (HCWs) is an important component of pandemic preparedness [7]. In particular, vaccination of this occupational group is important in order to protect the health and safety of the essential workforce and keep the healthcare system operating at maximum capacity during a pandemic [8,9]. In addition, the vaccination of HCWs is of vital importance given that they are used as a role model for their patients and the public [10]. The rapid development of effective vaccines against COVID-19 represents an extraordinary achievement, but it also fuels vaccine hesitancy [11,12]. Nevertheless, after the discovery and massive use of COVID-19 vaccines, research interests have shifted towards the study of vaccination coverage rather than vaccination acceptance. Plenty of data exist on the COVID-19 vaccination acceptance among health care workers [13,14,15], but there are very limited data on the prevalence and determinants of COVID-19 vaccination coverage among HCWs, and in particular among physicians.

Consequently, the aim of this study was to evaluate the coverage of COVID-19 vaccination and associated factors among physicians, members of the largest Greek Medical Association, Athens Medical Association.

## 2. Materials and Methods

### 2.1. Study Design and Participants

A descriptive, cross-sectional online study was conducted among the members of the Athens Medical Association (I.S.A.). The survey was conducted over the period 25 February to 13 March 2021. All members of I.S.A. were invited to participate in the anonymous online survey. A structured, anonymous questionnaire was used.

### 2.2. Questionnaire

The questionnaire included questions on demographics (sex, age, and occupational characteristics), perceptions of the importance of vaccinations, attitudes towards, safety and effectiveness of vaccines. In addition, the questionnaire included questions on COVID-19 (“Have you been vaccinated against COVID-19?” Answer options: Yes/No), and influenza vaccination coverage for flu season 2020–2021(“Have you been vaccinated with the influenza vaccine (season 2020–2021)?” Answer options: Yes/No). Vaccination coverage against COVID-19 included the receipt of one or two doses of the Pfizer–BioNTech. In the case of COVID-19 vaccination refusal, the participants were requested to report the reason of non-vaccination (pending vaccination appointment; not at risk of COVID-19 disease; opposition to vaccinations; concerns over side effects). Subjects were asked to rate on a four-point Likert scale (answer options: fully agree, agree, disagree, and fully disagree), the importance, effectiveness, and safety of vaccinations, as well as possible concerns over COVID-19 vaccination side effects. Moreover, the respondents were asked to evaluate the quality of COVID-19 vaccine-related information from Greek public health authorities. Last, the participants were asked about their sources of information on the safety of COVID-19 vaccines. A secondary aim of our survey was to obtain information from physicians on the role of primary health care in the national response to the COVID-19 pandemic in Greece. The participants were asked to report their perceptions on the role of primary health care in the COVID-19 response in Greece (“The primary health care role was satisfactory in the COVID-19 response in Greece”; “The primary health care should take a more active role in the COVID-19 response”; “Strengthening primary health care can help decongest hospital units during COVID-19 pandemic”).

### 2.3. Statistical Analysis

All data were analyzed with SPSS software, version 25. Relative (%) and absolute frequencies were presented for qualitative variables, whereas quantitative variables were presented using mean ± standard deviation. Chi-squared test (χ^2^) was used for the univariate analysis of qualitative variables. Student’s t-test was used for the univariate analysis of continuous variables after assessment for normality with the Kolmogorov–Smirnov test. Variables with a *p* value < 0.25 in the univariate analysis were included in a stepwise binary logistic regression analysis model, in order to explore potential independent associations with COVID-19 vaccination coverage. Adjusted odds ratios (AORs) and 95% confidence intervals (C.I.) were calculated. The level of statistical significance level was set at *p* = 0.05. All analyses were performed using SPSS. 25 (IBM SPSS Statistics for Windows, Version 25.0. IBM Corp.: Armonk, NY, USA).

### 2.4. Ethics

The study was conducted according to the principles of the Declaration of Helsinki of 1975, as revised in 2008. The participants provided anonymous informed consent on the survey platform before they could proceed to the electronic completion of the questionnaire. All participants gave informed consent for participation without any monetary incentives being offered. The protocol of the study was approved by the Board of the Athens Medical Association (Code: 18.02.21).

## 3. Results

The total number of participants was 1993 of a possible 25,900 members of the Athens Medical Association (response rate = 8%). Among them, 1192 (59.8%) were male and 801 (40.2%) were female. The mean age was 52.9 years (SD = 10.73). (Table 1). The distribution of the employment status was as follows: 19.7% (*n* = 392) physicians working in the National Health System (NHS), 74.3% (*n* = 1481) working in the private sector, 2.9% (*n* = 57) working in university hospitals and 3.2% (*n* = 63) working with the Greek Army (Table 1). The reported vaccination coverage against COVID-19 was 85.3%. The main reasons of no vaccination (*n* = 292) were pending vaccination appointment (63.4%), followed by safety concerns (33.5%) (Table 2).

### 3.1. Univariate Analysis

Univariate analysis (Table 3) has shown that older age, perception that vaccines are safe, effective and important tools for the protection of public health, and fear over COVID-19 vaccine-related side effects, were significantly associated with COVID-19 vaccination. Moreover, perceived reliability of the COVID-19 vaccination information received from the Greek Public Health Authorities was associated with increased COVID-19 vaccination coverage. This is also the case with history of influenza vaccination (flu season 2020–2021), with physicians who reported flu vaccination to present higher vaccination coverage against COVID-19 in comparison to their colleagues who were unvaccinated against flu. Notably, physicians who were being informed about the COVID-19 vaccines by independent websites and social media recorded lower COVID-19 vaccination coverage than health workers who were being informed by other sources. There was no significant difference in the COVID-19 vaccination status by sex and type of employment.

### 3.2. Multivariate Analysis

Logistic regression analysis (Table 4) demonstrated that no fear over COVID-19 vaccination-related side effects (AOR = 3.17; 95% C.I. = 2.39–4.19), history of influenza vaccination for flu season 2020–2021 (AOR = 2.31; 95% C.I. = 1.74–3.07), perception that the information on COVID-19 vaccination from the national public health authorities is reliable (AOR = 2.2;95% C.I. = 1.62–2.99), and perception that vaccines in general are safe (AOR = 3.16; 95% C.I. = 1.12–8.9), were found to be independent predictors of reported COVID-19 vaccination coverage. Although COVID-19 vaccination coverage increased with age, the association was not significant in logistic regression analysis (AOR = 1.008; 95% C.I. = 0.99–1.02).

### 3.3. Role of the Primary Health Care (PHC)

The majority of participants (70%) reported that Primary Health Care (PHC) had an important role in the national fight against COVID-19. On the other hand, the vast majority (93%) of the members of ISA reported that PHC should take a more active role in the COVID-19 response. Last, the overwhelming majority of the participants (94%), believed that strengthening PHC would decongest hospital units in the context of the COVID-19 pandemic.

## 4. Discussion

We conducted a cross-sectional study of vaccination coverage among 1993 members of the largest medical association in Greece. Vaccination coverage in our study was 85.3%. It should be noted that the main reason for non-vaccination was a pending vaccination appointment and therefore the reported vaccination coverage may have been even higher. Notwithstanding this point, this vaccination rate was higher in comparison to previous data of COVID-19 vaccination acceptance among HCWs [16,17,18,19,20]. In particular, a year ago, a survey collecting health care professionals’ (HCP) views in Greece, estimated the uptake of a future vaccine against SARS-CoV-2 at 43% [16]. Subsequent studies showed improved vaccination acceptance ranged from 51.1% to 78.5% [17,18,19]. A large prospective cohort study among HCWs in the United Kingdom found that vaccination coverage was 89% [20]. A cross-sectional study among post-graduate residents and fellows in Texas, United States, reported that 95.1% of the participants were vaccinated [21]. Another study among HCWs of a teaching hospital in the United Kingdom reported 82.5% COVID-19 vaccination coverage by the end of February 2021 [22]. Vaccination hesitancy comprised three components: confidence, complacency, and convenience (access to vaccines). The major reason for non-vaccination was pending vaccination appointment (63%), and it is possibly related to the third component of hesitancy (convenience). Around a third (33.5%) of non-vaccinated participants expressed concerns about vaccine safety. Interestingly, half of them were concerned about the rapid pace of COVID-19 vaccine development. This finding corroborates early reports among Israeli health care workers and the general public which indicated fear over safety due to the rapid development of the vaccines [23]. Additionally, focusing on the findings from a general population nation-wide study, the prevailing reasons against COVID-19 vaccination were safety concerns related to the duration of clinical trials and potential side effects [24]. Τhis figure underlines the imperative need for educational initiatives for both health care workers and the general population in order to properly address these concerns. Furthermore, we found that respondents who received information on COVID-19 vaccines from social media had lower COVID-19 vaccination coverage. There is research evidence that users of social media present lower vaccination coverage regarding influenza and lower human papillomavirus (HPV) vaccination acceptance [24,25]. Social media can serve as a vehicle for the spread of COVID-19 vaccine misinformation, and therefore may promote vaccine hesitancy [26]. In particular, exposure to social media may increase vaccination-related perceived risk and decrease the perception of vaccination benefits [27]. Furthermore, there are policy implications for social media companies which should adopt effective and flexible policies to control vaccine-related misinformation.

Multivariate analysis showed that safety concerns related to both previous routine vaccination and COVID-19 vaccination were independent drivers of non-COVID-19 vaccination coverage among the physicians under study. This finding is plausible and in line with the results of a recently published systematic review, which reports that concerns about safety and side-effects were among the top reasons for COVID-19 vaccination hesitancy [14]. Logistic regression analysis revealed a strong, significant independent impact of trust to Greek public health authorities on COVID-19 vaccination acceptance. This finding complies well with previous research in Greece among health care workers and general population [19,24], and has policy implications. The reliability of the information related to COVID-19 vaccination provided by the Greek public health authorities is an opportunity which should be broadly exploited by policymakers in order to combat vaccination hesitancy and further improve COVID-19 vaccination coverage among HCWs and the general population. Influenza vaccination (flu season 2020–2021) was found to be a significant predictor of COVID-19 vaccination coverage. This finding correlates well with the results of a cross-sectional study among HCWs from France and French-speaking parts of Belgium and Canada, which reported an independent effect of influenza vaccination (season 2019–2020) on the acceptability of COVID-19 vaccination [28]. Other possible interpretations may include the increased prevalence of physicians with comorbidities among those vaccinated with influenza vaccine, and perceptions that flu vaccination could be protective against COVID-19 [29,30]. However, the cross-sectional design of the present study does not allow us to definitely conclude if influenza vaccination is a determinant of COVID-19 vaccination or vice versa.

Knowledge, emotions, and cultural perceptions appear as determinants that shape one’s intention to get vaccinated. From this long list of variables assessed, it is surprising that primary care service experience does not appear among those influencing vaccination figures. It would be interesting to include and examine primary care service provision or delivery as variables influencing the intention to get vaccinated. Likewise, it would be valuable to see how primary care physicians and attendees interact and deal with vaccination propensity since the primary care environment can blend facets of knowledge, behavior, culture and health in what is called personalized care. It is also known that patients feel comfortable receiving medical care from providers who are culturally compatible to understand their lived experiences [31]. For those who are worried or resilient, primary care is the ideal font of consistent and truthful information [31].

Thus, a secondary aim of our survey was to obtain information from physicians on the role of primary health care in the national response to the COVID-19 pandemic in Greece. Early this year, Hardnen et al. highlighted the extended delivery option through primary care by providing a more personalized approach and being more efficacious in vaccinating vulnerable groups across the country [32]. Trust and experience in delivery of the routine vaccines supported the argument that primary care has a pivotal advantage to overcome obstacles and make the program successful [32]. Many national health systems have found ways to increase their labor force in primary care [33]. The engagement of primary care in the vaccination mission has had a diverse impact across different countries, although the main workforce has usually been skilled and from state public health departments [33].

As of 16 September 2021, in Greece there have been 622,761 reported cases of COVID-19, and 14,354 deaths. In addition, 3077 patients have been discharged from intensive care units [34]. Interestingly, two-thirds of the participants reported that PHC played an important role in the national fight against COVID-19. In fact, it is expected that the current network of primary health care may have reduced the burden of COVID-19 cases in hospitals. However, we are unable to quantify the burden of COVID-19 hospital admissions which have been avoided due to the role of primary health care. The large majority of participants (94%), reported that strengthening PHC would further decongest hospital units in the context of the COVID-19 pandemic. This finding provides policymakers with evidence in order to strengthen the primary healthcare system and create a robust system of PHC.

Additionally, academic dialogue and primary care experience exchange on issues such as health professional empowerment may help to enhance inter-professional training and actions throughout the entire health system [35], and to promote work in synergism with community resources. If inertia prevails, vaccination hesitancy may convert personal uncertainty to a collective rupture. Collaboration of all healthcare providers is welcome [36], in order to deliver strong social snowball messages and preserve resources in a pandemic that still shows signs of resistance after months of human efforts. 

Our results are subject to several limitations to be considered prior to the interpretation of the results. First, the cross-sectional nature of the study could not enable us to infer causation between risk factors investigated and the outcome (COVID-19 vaccination coverage). Second, our study was questionnaire-based and there is a potential for information bias to occur. Third, we acknowledge the low response rate of the present survey, and since we were not able to obtain responses from non-respondents, this may be a source of selection bias. Nevertheless, it should be noted that a low response rate of online surveys has been a concern for many scholars and researchers [37]. In addition, our study was conducted during COVID-19 pandemic conditions, and it was very difficult for us to conduct an in-person survey. Furthermore, there is some evidence that the prevalence of self-reported health variables in public health studies may be underestimated due to selective non-response effects [38]. Nevertheless, key findings of our study (e.g., positive association between exposure to information received from Greek public health authorities and COVID-19 reported vaccination coverage, and the impact of fear of side effects on vaccination coverage) are in line with the results of a previous study in Central Greece which presented a twofold increased response rate in comparison to the present study. Consequently, selection bias may have occurred, but it seems unlikely that this bias considerably affected the results of our study.

## 5. Conclusions

We report a high vaccination coverage among Greek physicians, and an improvement in COVID-19 vaccination uptake in comparison to previous studies. The finding that participants reported high reliability of the information related to COVID-19 vaccination provided by the Greek public health authorities is an opportunity which should be broadly exploited by policymakers in order to combat vaccination hesitancy, and further improve COVID-19 vaccination uptake among HCWs/physicians and the general population. Last, our results provide policymakers with evidence to strengthen the primary healthcare system in Greece.

## Figures and Tables

**Table 1 vaccines-09-01134-t001:** Description of participants.

		*N*	%
Gender	Male	1192	59.8
	Female	801	40.2
Distribution of employment	Doctors working in the NHS	392	19.7
	Doctors working in the private sector	1481	74.3
	Doctors working in universities	57	2.9
	Doctors working with the Greek Army	63	3.1
Age (years)		52.9 ± 10.7	

**Table 2 vaccines-09-01134-t002:** Reasons for reported COVID-19 non-vaccination.

Reasons	*N*	%
Pending vaccination appointment	185	63.4
I am not at risk of COVID-19 disease	6	2.1
I am opposed to vaccinations	3	1.0
The time of the development of the vaccines was short	50	17.1
Fear of side effects	48	16.4
Total	292	100.0

**Table 3 vaccines-09-01134-t003:** Univariate analysis of COVID-19 vaccination coverage.

Variable	COVID-19 Vaccination Coverage		
	Yes (%)	No (%)	*p* Value
Sex			0.102
Male	1030 (86.4)	162 (13.6)	
Female	671 (83,8)	130 (16.2)	
Age (Years, Mean, SD)	53.2 ± 10.7	51.1 ± 10.9	0.002
Employment status			0.383
Working in the public sector (NHS, Army, Universities)	443 (86.5)	69 (13.5)	
Working in the private sector	1258 (84.9)	223 (15.1)	
The vaccines are important for public hHealth			0.001
Fully Agree/Agree	1698 (85)	288 (15)	
Fully disagree/Disagree	3 (43)	4 (57)	
In general, vaccines are safe.			<0.001
Fully Agree/Agree	1693 (86.4)	267 (13.6)	
Fully disagree/Disagree	8 (24.2)	25 (75.8)	
In general, vaccines are effective.			<0.001
Fully Agree/Agree	1696 (86)	278 (14)	
Fully disagree/Disagree	5 (26)	14 (74)	
I am concerned over COVID-19 vaccination side effects.			<0.001
Fully Agree/Agree	507 (72.7)	190 (27.3)	
Fully disagree/Disagree	1194 (92.1)	102(7.9)	
The information I have received on vaccination against COVID-19 from the Greek Public Health authorities is reliable.			<0.001
Fully Agree/Agree	1453 (89.3)	175 (10.7)	
Fully Disagree/Disagree	248 (67.9)	117 (32.1)	
Have you been vaccinated with the influenza vaccine 2020–2021?			<0.001
Yes	1361 (89.4)	161 (10.6)	
No	340 (72.2)	131 (27.8)	
Source of information on COVID-19 vaccines			0.001
Biomedical scientific publications, International Health Organizations, website of the Greek CDC (EODY), website of the medical association of Athens, television/radio/newspapers	1582 (86.2)	254 (13.8)	
Independent websites and social media	119 (75.8)	38 (24.2)	

**Table 4 vaccines-09-01134-t004:** Logistic regression analysis of COVID-19 vaccination coverage.

Independent Variable	AOR	95% C.I.	*p* Value
Reliable information from Greek Public Health Authorities			
Yes	2.20	1.62–2.99	
No	1.00 (ref)		<0.001
Fear of COVID-19 vaccine side effects			
No	3.17	2.39–4.19	<0.001
Yes	1.00 (ref)		
Influenza vaccination			<0.001
Yes	2.31	1.74–3.07	
No	1.00 (ref)		
In general, vaccines are safe			
Yes	3.16	1.12–8.9	
No	1.00 (ref)		0.036

## Data Availability

The study data are available from the corresponding author on reasonable request.

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
