# Peer review of "Reported COVID-19 Vaccination Coverage and Associated Factors among Members of Athens Medical Association: Results from a Cross-Sectional Study"

_vaccines, 2021, doi:10.3390/vaccines9101134_

Round 1

Reviewer 1 Report

This paper deals with an interesting and current topic. However, the very low response to the questionnaire (about 8%) makes the risk of a selection bias high and the results that have emerged are not very exportable. Moreover the authors do not comment on this problem and the study strong limitations in the discussion tand conclusions. It seems that, in this form and with limited response rate the survey cannot be accepted.

Author Response

Comments and Suggestions for Authors

This paper deals with an interesting and current topic. However, the very low response to the questionnaire (about 8%) makes the risk of a selection bias high and the results that have emerged are not very exportable. Moreover the authors do not comment on this problem and the study strong limitations in the discussion and conclusions. It seems that, in this form and with limited response rate the survey cannot be accepted.

Response: We would like to thank the reviewer for helpful and constructive comments. In fact, we have acknowledged the risk of selection bias due to the low response rate. However, as we commended in the revised form, low response rate is a well-known limitation in web based studies. The following paragraph has been included in the discussion section of the revised manuscript: “Last, we acknowledge the low response rate of the present study. Nevertheless, it should be noted that a low response rate of online surveys has been a concern for many scholars and researchers [36]. In addition, our study was conducted during COVID-19 pandemic conditions, and it was very difficult for us to conduct an in person survey”.

Reviewer 2 Report

In a cross-sectional questoinnaire-based study, the authors investigated COVID-19 vaccination coverage and associated factors among 1993 physicians who were all members of Athens Medical Association. They found a reported vaccination coverage of 85.3% and identified independent predictors of reported vaccination coverage including no fear over COVID-19 vaccination-related side effect, history of influenza vaccination and perception that the information from the national public health authories is reliable. Although this is an interesting study, I recommend to improve the following points:

  1. As this was only a questionnaire-based study the authors should change "vaccination coverage" to "reported vaccination coverage" throughout the whole manuscript (i.e. line 27, title of table 3, line 124, line 261)
  2. As the aim of this study was to evaluate coverage of COVID-19 vaccination among physician, I recommend that the authors stick to the term physician or medical doctor, but do not use the term health care workers or health workers (i.e. line 24, line 30, line 139). As healthcare workers do not only include physicians, but also nurses, emergency medical personnel, dental professionals and medical students. It is not known from a lot of other studys that vaccination rates can differ among these groups.
  3. The tables need to be optimized. First, the tables should be consecutively numbered and ordered in the manuscript. Second, in Table 1-3 for the decimal point of numbers use a point instead of a comma. Third, the style of Table 3 needs to be revised, as results of columns (absolute numbers and precentages) are shifted against each other and for the variable age and the column No COVID-19 vaccination coverage +/- is missing
  4. Concerning the statistical analysis: as the authors perform a multivariable logistic regression model, they do not show univariate odds ratios, but adjusted odds ratios (i.e. line 108, line 143-159, table 4). Moreover, as the number of variables is limited in this study, I recommend to include all variables in the model and do not use a stepwise model if not necessary. For the variables "reliable information from Greek Public Health Authorities" and "In general vaccines are safe" I recommend to change the reference from Yes to No, then the authors would receive positive predictors for all variables of Table 4.
  5. In the discussion section I recommend to delete the following parts:
    -Line 172-173:"The non vaccination rate in our survey was 14.7%". This is only a repetition/rewording of Line 163, but the authors could discuss that the main reason for non-vaccination was a pending vaccination appointment and therefore the reported vaccination coverage might have even been higher (94.6% instead of 85.3%)
    -Line 211 to 216: The authors should not support the perception that flu vaccination protects against COVID-19
  6. In the discussion section line 241 the authors discuss that they were unable to quantify the burden of COVID-19 hospital admissions with have been avoided due to the role of primary care. At least in the introduction they could include reported and excess COVID-19 death and hospital resources including ICU beds in Greece as published by https://covid19.healthdata.org/greece
  7. I recommend to adapt the first sentence of the conclusion as this study did not compare two different study periods.

Author Response

Reviewer 2

In a cross-sectional questionnaire-based study, the authors investigated COVID-19 vaccination coverage and associated factors among 1993 physicians who were all members of Athens Medical Association. They found a reported vaccination coverage of 85.3% and identified independent predictors of reported vaccination coverage including no fear over COVID-19 vaccination-related side effect, history of influenza vaccination and perception that the information from the national public health authorities is reliable. Although this is an interesting study, I recommend to improve the following points:

As this was only a questionnaire-based study the authors should change "vaccination coverage" to "reported vaccination coverage" throughout the whole manuscript (i.e. line 27, title of table 3, line 124, line 261)

Response: We would like to thank the reviewer for helpful and constructive comments. We have revised the text, accordingly.

As the aim of this study was to evaluate coverage of COVID-19 vaccination among physician, I recommend that the authors stick to the term physician or medical doctor, but do not use the term health care workers or health workers (i.e. line 24, line 30, line 139). As healthcare workers do not only include physicians, but also nurses, emergency medical personnel, dental professionals and medical students. It is not known from a lot of other studies that vaccination rates can differ among these groups.

Response: We modified the text, accordingly.

The tables need to be optimized. First, the tables should be consecutively numbered and ordered in the manuscript. Second, in Table 1-3 for the decimal point of numbers use a point instead of a comma. Third, the style of Table 3 needs to be revised, as results of columns (absolute numbers and percentages) are shifted against each other and for the variable age and the column No COVID-19 vaccination coverage +/- is missing

Response: We would like to thank the reviewer for the helpful comment. We have corrected the tables, accordingly.

Concerning the statistical analysis: as the authors perform a multivariable logistic regression model, they do not show univariate odds ratios, but adjusted odds ratios (i.e. line 108, line 143-159, table 4).

 Response: We have added the term Adjusted Odds Ratio on to table 4.

Moreover, as the number of variables is limited in this study, I recommend to include all variables in the model and do not use a stepwise model if not necessary.

Response: In fact all variables with p value <0.25, were included in the final model but only statistically significant results were presented.

 For the variables "reliable information from Greek Public Health Authorities" and "In general vaccines are safe" I recommend to change the reference from Yes to No, then the authors would receive positive predictors for all variables of Table 4.

Response: Thank you for this comment. Table 4 has been modified, accordingly.

  In the discussion section I recommend to delete the following parts:

-Line 172-173:"The non-vaccination rate in our survey was 14.7%". This is only a repetition/rewording of Line 163, but the authors could discuss that the main reason for non-vaccination was a pending vaccination appointment and therefore the reported vaccination coverage might have even been higher (94.6% instead of 85.3%)

Response: We have amended the text, accordingly.

-Line 211 to 216: The authors should not support the perception that flu vaccination protects against COVID-19

Response: We would like to thank the reviewer for gave us the opportunity to clarify our position.  We didn’t support the perception that influenza vaccination protects against COVID-19. Simply, we tried to e speculate on the association between flu vaccination and COVID-19 vaccination.

In the discussion section line 241 the authors discuss that they were unable to quantify the burden of COVID-19 hospital admissions with have been avoided due to the role of primary care. At least in the introduction they could include reported and excess COVID-19 death and hospital resources including ICU beds in Greece as published by https://covid19.healthdata.org/greece

Response: We have deleted the relative sentence.

I recommend to adapt the first sentence of the conclusion as this study did not compare two different study periods.

Response: We have made the necessary adaptations.

Round 2

Reviewer 2 Report

The authors only partly followed the reviewers' recommendation:

Table 3 is still mentioned first in the text. There are still mistakes in Table 1-3 (59,9 +/- 10,7; 100,00; 83,8,...).  Although the reference was changed for the variables "reliable information from Greek Public Health Authorities" and "In general vaccines are safe" in Table 4, the main text Page 7 (line 174, 176) still shows the old numbers.
At least in the introduction the authors should show some numbers concerning the pandemic situation in Greece (e.g. excess COVID-19 death and hospital resources including ICU beds).

Author Response

The authors only partly followed the reviewers' recommendation:

Table 3 is still mentioned first in the text. There are still mistakes in Table 1-3 (59,9 +/- 10,7; 100,00; 83,8,...).  Although the reference was changed for the variables "reliable information from Greek Public Health Authorities" and "In general vaccines are safe" in Table 4, the main text Page 7 (line 174, 176) still shows the old numbers.

Response: We would like to thank the reviewer again. We have corrected the presentation of all the tables, accordingly. 

At least in the introduction the authors should show some numbers concerning the pandemic situation in Greece (e.g. excess COVID-19 death and hospital resources including ICU beds).

Response: We have revised the discussion section (page 9) of the manuscript as follows: “As of September 16, 2021, in Greece there have been 622761 reported cases of COVID 19, and 14354 deaths. While 3077 patients exited Internal Care Units [34].   Interestingly, two-thirds of the participants reported that PHC played an important role in the national fight against Covid-19. In fact, it is expected that the current network of primary health care reduced burden of COVID-19 cases in hospitals. However, we are unable to quantify the burden of COVID-19 hospital admissions which have been avoided due to the role of primary health care”.
